

# 17β-estradiol upregulates oxytocin and the oxytocin receptor in C2C12 myotubes

Enrica Berio, Sara Divari, Laura Starvaggi Cucuzza, Bartolomeo Biolatti and Francesca Tiziana Cannizzo

Department of Veterinary Science, University of Turin, Grugliasco, Torino, Italy

## ABSTRACT

**Background**. The endocrinology of skeletal muscle is highly complex and many issues about hormone action in skeletal muscle are still unresolved. Aim of the work is to improve our knowledge on the relationship between skeletal muscle and 17β-estradiol.

**Methods**. The skeletal muscle cell line C2C12 was treated with 17β-estradiol, the oxytocin peptide and a combination of the two hormones. The mRNA levels of myogenic regulatory factors, myosin heavy chain, oxytocin, oxytocin receptor and adipogenic factors were analysed in C2C12 myotubes.

**Results**. It was demonstrated that C2C12 myoblasts and myotubes express oxytocin and its receptor, in particular the receptor levels physiologically increase in differentiated myotubes. Myotubes treated with 17β-estradiol overexpressed oxytocin and oxytocin receptor genes by approximately 3- and 29-fold, respectively. A decrease in the expression of fatty acid binding protein 4 (0.62-fold), a fat metabolism-associated gene, was observed in oxytocin-treated myotubes. On the contrary, fatty acid binding protein 4 was upregulated (2.66-fold) after the administration of the combination of 17β-estradiol and oxytocin. 17β-estradiol regulates oxytocin and its receptor in skeletal muscle cells and they act in a synergic way on fatty acid metabolism.

**Discussion**. Oxytocin and its receptor are physiologically regulated along differentiation. 17β-estradiol regulates oxytocin and its receptor in skeletal muscle cells. 17β-estradiol and oxytocin act in a synergic way on fatty acid metabolism. A better understanding of the regulation of skeletal muscle homeostasis by estrogens and oxytocin peptide could contribute to increase our knowledge of muscle and its metabolism.

Corresponding author
Sara Divari, sara.divari@unito.it

## INTRODUCTION

Hormones such as estrogens, testosterone, growth hormone, insulin, insulin-like growth factor-I and glucocorticoids have a profound influence on skeletal muscle and they are important regulators of the remodelling process. Anabolic hormones stimulate muscle growth in animals and humans by increasing protein synthesis, by decreasing protein breakdown or both. While the anabolic effects of androgens are well known (*Dubois et al., 2012*), the effects of estrogens on skeletal muscle anabolism have only been discovered

recently (*Greising et al., 2009*), in particular estrogens have been shown to stimulate muscle repair and regenerative processes (*Enns & Tiidus, 2010*).

Skeletal muscle myogenesis is controlled by myogenic regulatory factors (MRFs), such as myogenic differentiation 1 (MYOD), myogenic factor 5 (MYF5), myogenin (MYOG) and myogenic factor 6 (MYF6), and mature skeletal muscle expresses structural muscle proteins, such as myosin heavy chain (MYH) and tropomyosin (*Charbonnier et al., 2002*). MRFs are expressed in a time-dependent manner and regulate differentiation of muscular cells, in particular they control the fusion of myoblasts (MBs) (mononucleated muscle precursor) into multinucleated cells called myotubes (MTs) (*Dedieu et al., 2002*).

Growth-promoting agents are purported to increase skeletal muscle fiber size and they could be illegally used to improve sport performances or meat production. Previous studies described the effect of 17$\beta$-estradiol (E2) on the gene expression levels in skeletal muscle of veal calves. In particular, *De Jager et al. (2011)* and *Divari et al. (2013)* demonstrated a strong increase in the expression of the oxytocin (*Oxt*) precursor and oxytocin receptor (*Oxtr*) genes in the skeletal muscle of E2-treated veal calves and an intense raise of the plasmatic concentration of circulating oxytocin peptide (OXT). The high serum OXT concentration was likely due to the estrogen influence on OXT production within the supraoptic and paraventricular nuclei of the hypothalamus and/or its release from the posterior pituitary into the blood stream (*Chung, McCabe & Pfaff, 1991*; *Nomura et al., 2002*). Moreover, *Oxt* precursor gene is increased in bovine skeletal muscle in the late stages of foetal development (*De Jager et al., 2011*), suggesting that OXT could have a role in muscle growth during both foetal and postnatal development in animals (*De Jager et al., 2011*; *Divari et al., 2013*) and that E2 can have a regulatory effect on OXT production in skeletal muscle.

In mammals OXT is mainly synthesized in the central nervous system and it induces uterine contractions during parturition and milk ejection during lactation. In recent years, the classic concept of OXT action has been greatly expanded because of the discovery of novel sites of OXT and OXT receptor (OXTR) gene and protein expression such as the testes, ovaries, heart and lungs (*Assinder et al., 2000*; *Jankowski et al., 2004*; *Kiss & Mikkelsen, 2005*; *Péqueux et al., 2005*; *Zingg & Laporte, 2003*). Moreover, E2 was demonstrated to induce OXT/OXTR overexpression in several of those (*Feng et al., 2009*; *Sharma, Handa & Uht, 2012*). Many studies demonstrated that OXT has various metabolic effects in particular on muscular metabolism and regeneration, glucose metabolism, lipid profile and insulin sensitivity. In skeletal muscle OXT exerts its activity promoting glucose uptake and inducing lypolisis (*Elabd et al., 2014*; *Elabd & Sabry, 2015*). Similarly, E2 is intensively studied for its influence on energy balance, skeletal muscle and adipose tissue metabolism and glucose uptake regulation, but its role is still partially unclear (*Ropero et al., 2008*). In this regard, an important role on cell metabolism is played by fatty acid binding proteins 3 and 4 (FABP3 and 4) and peroxisome proliferator-activated receptor gamma (PPARG) which mediate fatty acid translocations, regulate cholesterol metabolism, participate to several signalling cascades (*Makowski & Hotamisligil, 2004*) and control adipogenesis (*Jeong & Yoon, 2011*).

Aim of the work was to evaluate the physiological expression of OXT and OXTR in MBs and MTs and to asses a relationship between estrogen administration and OXT/OXTR

expression in a reproducible model of skeletal muscle. In particular, it was verified if E2 and OXT could influence myofibers metabolism gene regulation. For this purpose, the C2C12 cell line was treated with E2, OXT or a combination of both molecules to assess the regulation of myogenic regulatory factors genes (*MRFs*), myosin heavy chain (*Myh)* and some factors involved in adipogenesis, such *Fabp3*, *Fabp4* and *Pparg*.

## MATERIALS AND METHODS

### C2C12 cell culture

C2C12 murine myoblasts (MBs) (European Collection of Cell Cultures, ECACC No. 91031101, Salisbury, UK) were grown in Dulbecco's modified Eagle's medium (DMEM; Sigma, St. Louis, MO, USA) supplemented with 10% heat-inactivated foetal bovine serum, 2 mM glutamine (Sigma) and 1% antibiotic antimycotic solution (Sigma). The cultured cells were maintained in 5% $CO_2$ atmosphere at 37 °C. To obtain differentiated myotubes (MTs), cells at 80% confluence were switched into differentiation medium (DM) containing 10% heat-inactivated horse serum.

Phenol red-free medium was used throughout the experiments to avoid its weak estrogen-like activity.

### BT474 cell culture

BT474 (American Type Culture Collection, ATCC®, HTB-20, Manassas, VA, USA) human mammary gland carcinoma cells were grown in DMEM supplemented with 10% heat-inactivated foetal bovine serum, 2 mM glutamine (Sigma) and 1% antibiotic antimycotic solution (Sigma) as positive control for the enzyme immunoassay (EIA). BT474 cells were grown in 100 mm petri dishes until 80% confluence, were starved and treated with E2 (10 nM) for 48 h to induce OXT secretion as described by *Cassoni et al. (2006)*.

### Experimental design

In all experiments, the sera were stripped with charcoal and dextran as described in *Sharma, Handa & Uht (2012)* to remove hormones.

To assess the physiological OXT and OXTR expression level, C2C12 MBs and MTs were respectively collected at day 3 and day 7. Subsequently, trials of acute and chronic administration of E2 were conducted on MTs to study gene expression modifications.

For the acute treatment trial, MTs were grown to day 6 and then starved with media containing stripped horse serum for 24 h before the E2 treatment (Sigma). The MTs were incubated with E2 (10 nM) at day 7 and then harvested at 0.5, 1, 2, 3, and 4 h of incubation (adapted from *Sharma, Handa & Uht (2012)*) to determine the *Oxt* and *Oxtr* expression levels. The control cultures were administered vehicle (Fig. 1).

For the chronic treatment trials, cells were treated with E2 (10 nM) or OXT peptide (10 µM) (Sigma), or a combination of E2 with OXT peptide (E2-OXT) (10 nM and 10 µM, respectively) every 48 h during differentiation from MB to MT (Fig. 1). OXT peptide was dissolved in 10 nM citrate buffer to increase its stability at high temperature (*Avanti et al., 2011*). Day 7 was chosen to harvest the differentiated MTs and to conduct the transcriptomic analysis because spontaneous contraction of MTs was observed, which

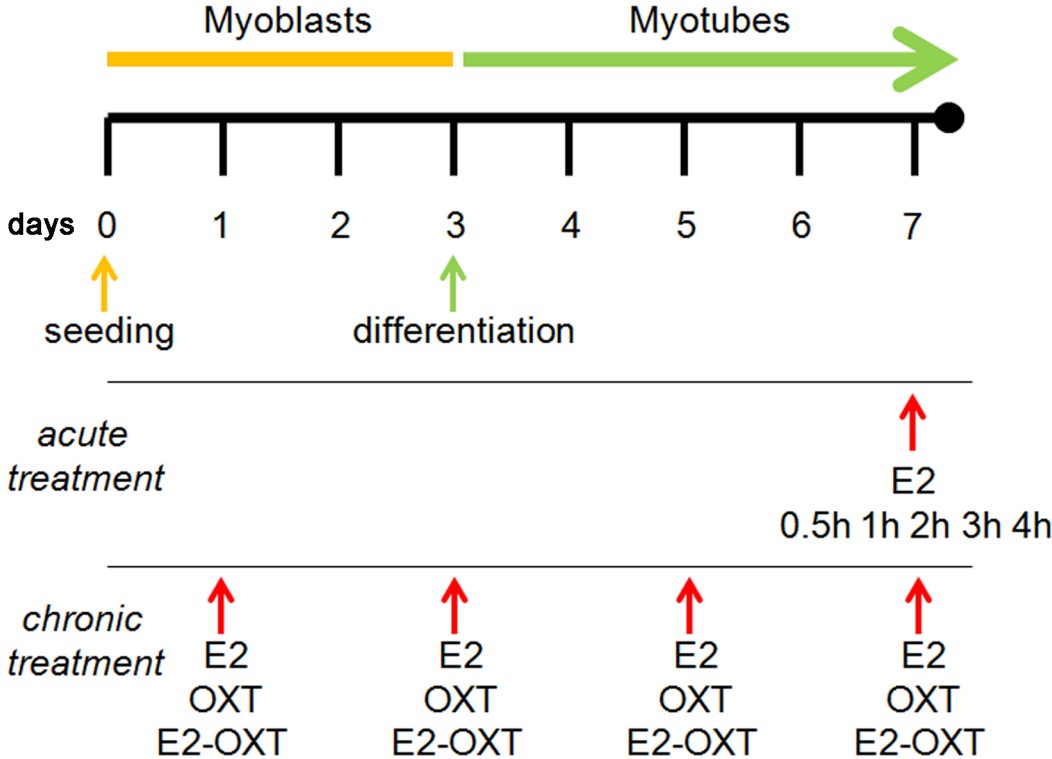

**Figure 1 Experimental scheme.** The C2C12 MBs were maintained in growth medium for 3 days (80–90% confluence). Myogenic differentiation was then induced from day 3 to day 7 by switching the cells to differentiation medium (DM). For the experiments involving acute treatment at day 7, the MTs were incubated with E2 (10 nM) for 0.5, 1, 2, 3, and 4 h. For the experiments involving chronic treatment, the cells were treated with E2 (10 nM), OXT (10 μM), or combination of E2 and OXT (10 nM and 10 μM, respectively) every 48 h during differentiation from MBs to MTs. The cells were harvested for qPCR analysis and western blotting.

is commonly associated with the maturation of MTs in culture. The levels of the *Oxt, Oxtr, Myf6, Myf5, Myod, Myog Myh, Fabp3, Fabp4* and *Pparg* mRNAs were determined. Each experiment was performed five times in duplicate.

## Myotube fusion index

The fusion index is a frequently used index of muscle cell differentiation and provides a measure of the proportion of the total cell populace that has fused (*Agley et al., 2012*). The fusion rate of the MTs that were chronically treated with E2, OXT and E2-OXT was evaluated at day 7 and compared to the untreated cell culture. The cells were washed with PBS and fixed in 10% neutral buffered formalin for 10 min. The fixed MTs were rinsed twice with PBS and stained with haematoxilin and eosin (H&E). Five control and five treated wells from independent experiments were analysed. For each well, the nuclei within the MTs and the nuclei of the unfused cells were counted in 10 randomly selected fields (200×) with Image Pro® plus (Version 3.0.1) software. A cell containing three or more nuclei was considered to be a MT (*Ge, Yu & Jiang, 2012*). For each slide, the fusion index percentage was calculated as (the number of fused nuclei/total nuclei)*100.

**Table 1 Primer sequences for qPCR.**

| Gene (RefSeq ID) | Forward primer (5′–3′) | Reverse primer (5′–3′) | Amplicon size (bp) |
|---|---|---|---|
| Oxt (NM_011025) | TGGCTTACTGGCTCTGACCT | GAGACACTTGCGCATATCCA | 94 |
| Oxtr (NM_001081147) | GCACGGGTCAGTAGTGTCAA | CCACATCTGCACGAAGAAGA | 120 |
| Myod (NM_010866) | TACAGTGGCGACTCAGATGC | TAGTAGGCGGTGTCGTAGCC | 116 |
| Myf5 (NM_008656) | CTGTCTGGTCCCGAAAGAAC | AAGCAATCCAAGCTGGACAC | 103 |
| Myog ( NM_031189) | CGATCTCCGCTACAGAGGC | GTTGGGACCGAACTCCAGT | 115 |
| Myf6 (NM_008657) | GGCTGGATCAGCAAGAGAAG | AGGAAATCCGCACCCTCA | 91 |
| Myh (NM_030679) | CGCAAGAATGTTCTCAGGCT | GCCAGGTTGACATTGGATTG | 110 |
| Fabp3 (NM_010174) | TAGGGAGCTAGTTGACGGGAA | ACGCCTCCTTCTCATAAGTCC | 83 |
| Fabp4 ( NM_024406) | GCAGAAGTGGGGATGGAAAGT | CTTGTGGAAGTCACGCCTTT | 96 |
| Pparg (NM_001127330) | CGAGTCTGTGGGGATAAAGC | TTCAATCGGATGGTTCTTCG | 92 |
| Ppia (NM_008907) | GCAAATGCTGGACCAAACAC | TCACCTTCCCAAAGACCACAT | 97 |

Notes.

Oxt, oxytocin gene; Oxtr, oxytocin receptor gene; Myod, myogenic differentiation 1 gene; Myf5, myogenic factor 5 gene; Myog, myogenin gene; Myf6, myogenic factor 6 gene; Myh, myosin heavy chain gene; Fabp3, fatty acid binding protein 3 gene; Fabp4, fatty acid binding protein 4 gene; Pparg, peroxisome proliferator-activated receptor gamma gene; Ppia, peptidylprolyl isomerase A.

## RNA extraction and relative quantification by qPCR

The total RNA from each cell culture sample was extracted using TRIzol reagent (Invitrogen, Life Technologies, Carlsbad, CA, USA), according to the manufacturer's protocol. The RNA concentration was determined by UV-Visible spectrophotometry and the RNA integrity was verified by automated gel electrophoresis system (Experion Instrument; Bio-Rad, Hercules, CA, USA). The cDNAs were synthesized from 1 μg of the total RNA using the QuantiTect Reverse Transcription Kit (Qiagen, Hilden, D), which included DNase reaction, according to the manufacturer's protocol.

To determine the relative amounts of specific Oxt, Oxtr, MRFs, Myh, Fabp3, Fabp4, and Pparg transcripts, the cDNAs were subjected to qPCR using the IQ5 detection system (Bio-Rad) and the IQ SYBR Green Supermix (Bio-Rad). The peptidylprolyl isomerase A (Ppia) cDNA was used as housekeeping gene control. The primer sequences were designed using Primer 3 (vers. 0.4.0) (Untergrasser et al., 2012) or Primer-BLAST (Ye et al., 2012), except for those for the Myh (Feng et al., 2009) and Ppia (Nishimura et al., 2008) genes, which were based on the literature (Table 1).

The levels of gene expression were calculated using the relative quantification assay based on the comparative $C_q$ method ($\Delta\Delta C_q$ method) (Bustin et al., 2009). The relative abundances of each transcript were recorded as $2^{-\Delta\Delta C_q}$ (fold increase) (Pfaffl, 2004).

Regarding the comparison of physiological Oxt and Oxtr expression in untreated MBs and MTs, the gene expression levels were described as mRNA arbitrary units ($2^{-\Delta C_q}$).

## Western blot analysis

The total proteins were extracted from cell lysate using RIPA buffer (50 mM Tris, pH 8.0, 150 mM NaCl, 1.0% IGEPAL CA-630, 0.5% sodium deoxycholate, 0.1% SDS and 2 mM EDTA) supplemented with a protease inhibitor cocktail (Sigma). The protein concentration was determined using the Bio-Rad DC Protein Assay. Proteins were boiled at 95 °C for 5 min in Leammli buffer and 35 μg of the total protein were resolved by

10% SDS–PAGE for OXTR assessment. The proteins were blotted to Hybond-P PVDF membrane (Amersham Biosciences, Piscataway, NJ, USA) using the Mini Trans-Blot cell (Bio-Rad). The blotted membranes were blocked with bløk-CH Buffer (Millipore KGaA, Darmstadt, D) for 1 h at room temperature, followed by overnight 4 °C incubation with the primary goat polyclonal anti-OXTR antibody (1:200; Santa Cruz Biotechnology). Proteins of rat heart was used as positive control (C+) for OXTR as other authors described (*Jankowski et al., 1998*). The membranes were subsequently incubated with a secondary horseradish peroxidase (HRP)-conjugated anti-goat antibody (1:10000), developed using the SuperSignal West Pico IgG Detection Kit (Thermo Fisher Scientific, Waltham, MA, USA) and signal was recorded on CL-XPosure X-ray film (Thermo Fisher Scientific) or by Chemi-Doc MP System (Bio-Rad). α-tubulin (1:10000, clone B-5-1-2; Sigma) was used as total protein loading control. Densitometric analysis of the bands was performed by the public domain ImageJ software (US National Institutes of Health, Bethesda, MD, USA; http://rsb.info.nih.gov/nih-image/), ($n = 3$).

### Enzyme immunoassay (EIA)

To study OXT peptide expression, the culture media from MBs and MTs, from the acutely treated MTs and BT474 were collected and immediately stored at −80 °C until further analysis. The soluble OXT peptide concentration was measured by enzyme immunoassay methodology using the EIA kit developed by Cayman Chemical (Tallinn, EE). The manufacturer reports a limit of detection (LOD) for this kit of approximately 18 pg/ml. OXT was extracted from 10 ml of culture medium aliquots using a solid-phase extraction (SPE), 1,000 mg C-18 Sep- Pack column (Supelco; Sigma-Aldrich, St. Louis, MO, USA) as previously described (*Divari et al., 2013*). Then, concentrated samples were collected in a glass tube and evaporated to dryness under a stream of nitrogen gas; they were reconstituted in 0.5 ml of assay buffer and immediately measured (concentrated 20-fold). The absorbance at 405 nm was read using Microplate Reader 680 Model (Bio-Rad) and a standard curve was created using a four parameter logistic function.

### Statistical analysis

All cell culture experiments were conducted in duplicate and were repeated five separate times.

All statistical analyses were performed using the GraphPad Prism 4 (vers. 4.03) software (GraphPad Inc., San Diego, CA, USA). Normal distribution was tested by the Kolmorov-Smirnov test. Grubbs' test was used to determine and exclude potential outliers. In the qPCR experiments in which results were expressed as $2^{-\Delta Cq}$, a paired $t$-test or Wilcoxon test were applied. For qPCR results expressed as differences in the relative expression ($2^{-\Delta\Delta Cq}$) of each target gene between the hormone-treated and untreated cells, data were analysed by ANOVA or Kruskal–Wallis for the acute trial, followed by Dunnett's or Dunn's post test, and by the paired $t$-test or Mann–Whitney test for chronic trial results. Data are presented as the means $\pm$ SEM from 5 experiments. $p < 0.05$ was considered significant.

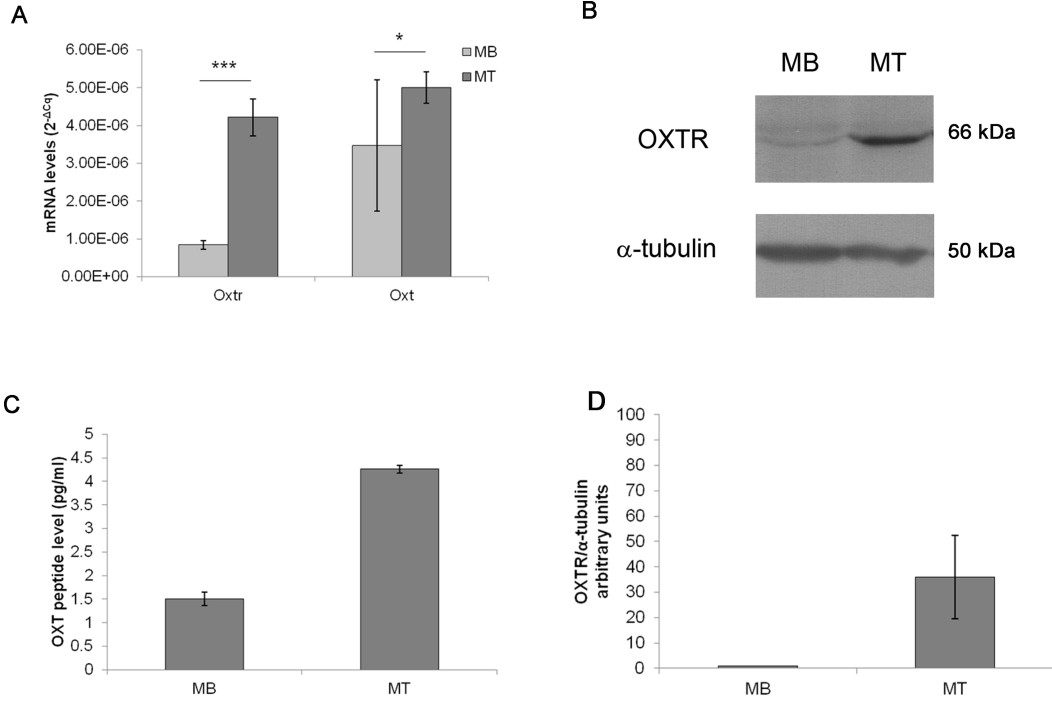

**Figure 2** ***Oxt and Oxtr* mRNA and protein levels in the control MB and MT cultures.** (A) The gene expression results are presented as the means ± SEM of *Oxt/Ppia* and *Oxtr/Ppia* mRNA level and expressed as $2^{-\Delta Cq}$. (B) Western blot of the OXTR protein in the control MB and MT lysates, (C) EIA test for OXT peptide concentration in culture growth medium expressed as pg/ml, (D) Densitometric analysis of western blot bands of OXTR. Band intensity was normalized versus the loading control and presented as arbitrary units.

## RESULTS

### Basal levels of OXT and OXTR expression during differentiation of C2C12 MBs into MTs

The basal levels of *Oxt* mRNA expression were extremely low in MBs (3.47E−06 mRNA arbitrary units); while, during myogenic differentiation into MTs, the *Oxt* mRNA concentration increased (5.01E−06 mRNA arbitrary units, $p < 0.05$). Similarly, *Oxtr* mRNA expression in MBs was scarse (8.40E−07 mRNA arbitrary units) and increased in differentiated MTs (4.28E−06 mRNA arbitrary units, $p < 0.001$) (Fig. 2A). Protein expression of OXT in MBs and MTs was demonstrated by EIA test (1.50 pg/ml in MBs and 4.26 pg/ml in MTs respectively) (Fig. 2C). OXTR protein was demonstrated and quantified by western blot (Figs. 2B and 2D).

### Effect of acute E2 administration on OXT and OXTR expression in C2C12 MTs

In the acute assay, *Oxt* mRNA was upregulated by approximately 3-fold after 0.5 h of E2 incubation ($p < 0.01$) compared with the control cells. After 3 and 4 h of E2 treatment, the mRNA levels progressively decreased until control levels (Fig. 3A). *Oxtr* mRNA levels were not significantly regulated by treatment (Fig. 3B).

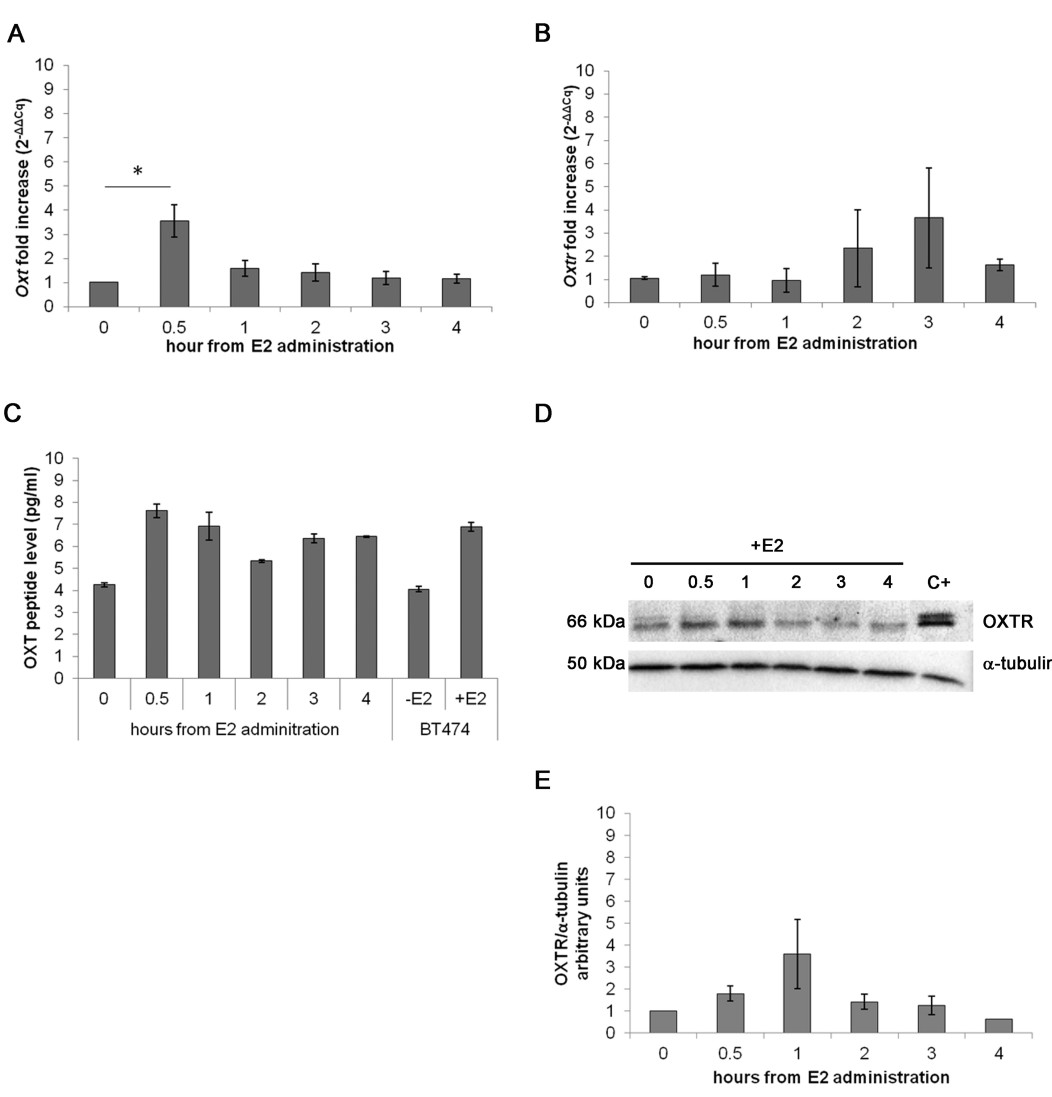

**Figure 3** **Effect of the acute E2 treatment on OXT and OXTR expression in the MTs.** (A) and (B) *Oxt* and *Oxtr* gene expression in the MTs respectivley. The results are presented as the means ± SEM of the *Oxt* fold increase expressed as $2^{-\Delta\Delta Cq}$. The samples were analysed in duplicate in five independent experiments (*$p < 0.05$), (C) OXT peptide concentration (pg/ml) in the culture media (10 ml) of the acute E2-treated MTs. Lysate from E2-treated BT474 cells (BT474 + E2) was used a positive control. The OXT peptide concentrations were determined by EIA Kit. (D) A representative western blot showing the expression of OXTR in the lysates of the acute E2-treated MTs. α-tubulin was used as loading control. Positive control (C+) (E) densitometric analysis of western blot bands of OXTR. Band intensity was normalized versus the loading control and presented as arbitrary units.

The EIA test on the growth media from the examined cell culture showed different OXT peptide concentration between samples: 7.61 pg/ml of OXT peptide were detected after 0.5 h of E2 administration against the 4.26 pg/ml of OXT of the control sample. The media from BT474 E2-treated cells was used as positive control for OXT secretion demonstrating a total OXT concentration of 6.90 pg/ml (Fig. 3C). OXTR protein expression was studied by Western blot (Figs. 3D and 3E) and no statistically significant differences were recorded.

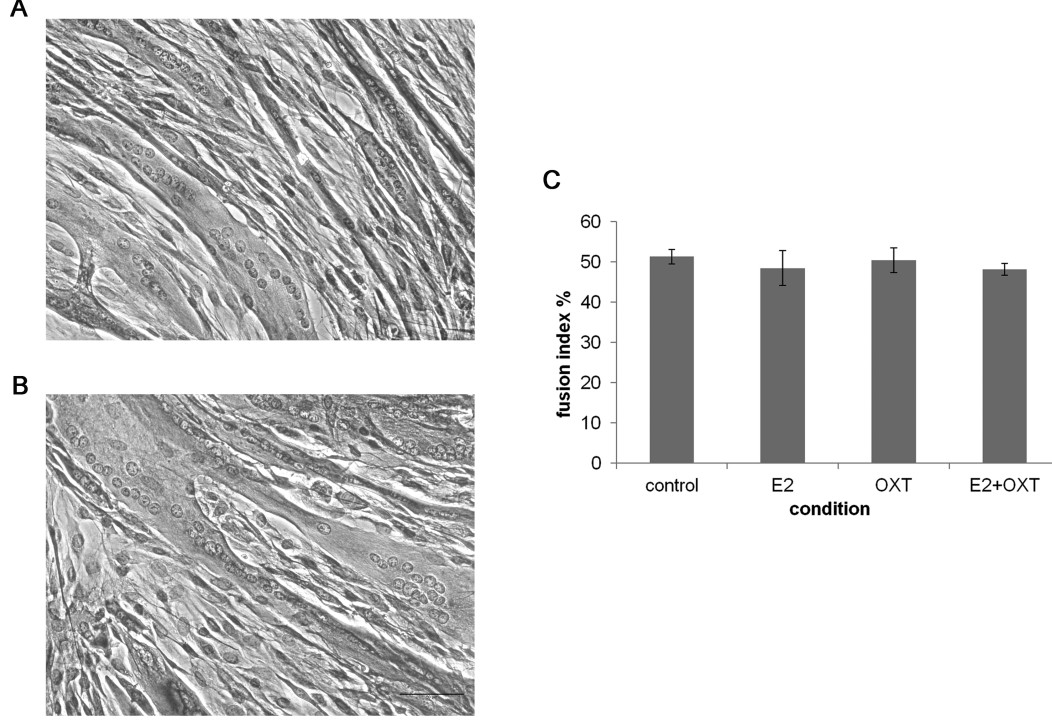

**Figure 4** **The fusion index.** (A) and (B) Representatives images of control culture and E2-treated MTs respectively (H&E, 50 μm bar), (C) The fusion index percentage of chronic E2-, OXT-, and E2–OXT-treated MTs was calculated as (the number of fused nuclei/total nuclei)*100.

## Effect of chronic hormone administration on MTs fusion index

To study the effect of chronic E2, E2-OXT and OXT treatments on myogenic development fusion index was studied in MTs and representative images are shown in Figs. 4A and 4B. The fusion index was not affected by chronic administration of E2 nor by OXT or the combination of both molecules (Fig. 4C).

## Effect of chronic hormone administration on *Oxt, Oxtr, MRFs, Myh, Fabp3, Fabp4* and *Pparg* gene expression in C2C12 MTs

The *MRF* mRNA levels in the C2C12 MTs were not significantly regulated by any of the chronic treatments (Figs. 5A–5C). A mild downregulation of the *Myh* gene was observed in the chronically treated cells; in particular E2 induced a significant decrease of approximately 0.90-fold compared with the control culture ($p < 0.05$) (Fig. 5A). Chronic treatments with E2-OXT and OXT did not significantly influence the expression of the studied genes involved in growth and differentiation of the MTs.

Chronic administration of E2 and E2-OXT induced a significant overexpression of the *Oxtr* gene. In particular, in the E2-treated cells the levels of the *Oxtr* mRNA were increased by approximately 29-fold compared with the control culture ($p < 0.05$) (Fig. 5D) and the combination of the two hormones induced this mRNA by approximately 11-fold compared with the control culture ($p < 0.05$) (Fig. 5F).

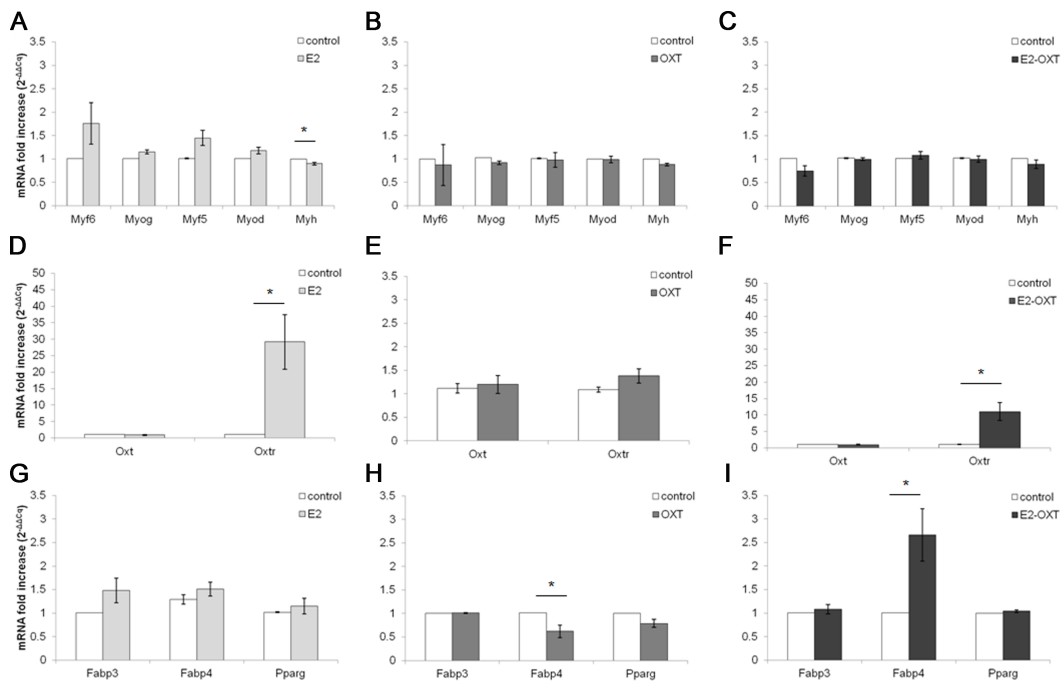

**Figure 5** **Effect of chronic hormone administration on *Oxt, Oxtr, MRFs, Myh, Fabp3, Fabp4* and *Pparg* gene expression in C2C12 MTs.** Gene expression of the several genes of interest was studied by qPCR. (A) Gene expression of *MRFs* genes and *Myh* in MTs treated with E2, (B) Gene expression of *MRFs* genes and *Myh* in MTs treated with OXT, (C) Gene expression of *MRFs* genes and *Myh* in MTs treated with E2-OXT, (D) Gene expression of *Oxt* and *Oxtr* genes in MTs treated with E2, (E) Gene expression of *Oxt* and *Oxtr* genes in MTs treated with OXT, (F) Gene expression of *Oxt* and *Oxtr* genes in MTs treated with E2-OXT, (G) Gene expression of *Fabp3, Fabp4* and *Pparg* genes in MTs treated with E2, (H) Gene expression of *Fabp3, Fabp4* and *Pparg* genes in MTs treated with OXT, (I) Gene expression of *Fabp3, Fabp4* and *Pparg* genes in MTs treated with E2-OXT. The results are presented as the means ± SEM of the *Oxt* fold increase expressed as $2^{-\Delta\Delta Cq}$. The samples were analysed in duplicate in five independent experiments (*$p < 0.05$). *Myf6*, myogenic factor 6 gene; *Myog*, myogenin gene; *Myf5*, myogenic factor 5 gene; *Myod*, myogenic differentiation 1 gene; *Myh*, myosin heavy chain gene; *Oxt*, oxytocin gene; *Oxtr*, oxytocin receptor gene; *Fabp3*, fatty acid binding protein 3 gene; *Fabp4*, fatty acid binding protein 4 gene; *Pparg*, peroxisome proliferator-activated receptor gamma gene.

Moreover, chronic administration of E2-OXT in MTs induced a significant overexpression of *Fabp4* (2.66-fold) (Fig. 5I), which is involved in fatty acid handling, while a decrease in the expression of *Fabp4* (0.62-fold) was observed in the OXT-treated MTs (Fig. 5H)

## DISCUSSION

In the present study the expressions of OXT and OXTR genes and proteins were studied in C2C12 cell line and the modification induced by E2 and/or OXT treatment were analyzed.

We demonstrated for the first time that both C2C12 MBs and MTs express OXT and OXTR, in particular MBs shown a mild expression level which increased in differentiated MTs. OXTR is physiological expressed in human MBs (*Breton et al., 2002*) and in bovine muscle both *Oxt* and *Oxtr* genes are expressed during muscle development (*De Jager et al., 2011*). Since we demonstrated that the OXTR expression increases along with MTs

differentiation, it is likely that the OXT and OXTR system is involved in skeletal muscle development (*Lee et al., 2008*).

Successively, the action of E2 on gene expression in C2C12 MTs was studied and we shown that acute and chronic treatments induced different alterations.

Acute E2 treatments were performed adapting the protocol and time points used by *Sharma, Handa & Uht (2012)*. In their work a hypothalamic neuronal cell line derived from embryonic mice treated with E2 shown an overexpression of *Oxt* mRNA level during the first hour of treatment and returned to baseline by 2 h. Similarly in C2C12 MTs, acute treatments with E2 determined the overexpression of *Oxt* mRNA suggesting that the expression of the *Oxt* gene is rapidly upregulated in MTs following E2 administration. We demonstrated that OXT is expressed in C2C12 MTs and that acute E2 administration can control its release outside of the cell. Indeed, a higher concentration of OXT in the growth medium from treated samples was demonstrated compared with control samples by EIA test. These data suggest that E2 may regulate OXT release from skeletal muscle cells. We hypothesize that skeletal muscle is able to secrete OXT under E2 stimuli as similarly occurs in osteoblasts. Indeed, E2 induces abundant OXT production in bone marrow osteoblasts, with the potential for exerting an anabolic effect on the skeleton (*Colaianni et al., 2012*) through an autocrine feed-forward OXT/OXTR loop in which estrogen induces OXT release from osteoblasts and then OXT acts upon osteoblastic OXTR to further amplify estrogen action.

The detection of a modest secretory activity in the treated MTs suggests that acute stimuli with E2 regulate OXT expression and release, activating an OXT/OXTR loop in the skeletal muscle cells, similarly to the osteoblasts. Therefore, OXT may be referred to as a novel myokine.

Very few data about OXT effect in regulating gene expression in cell culture are available in literature. In the present work, considered that our preliminary *in vitro* results shown an overexpression of *Oxtr* mRNA after E2 treatment, we performed combined chronic treatments of E2 and OXT, the natural ligand of OXTR, to investigate OXT effects in cell culture. Moreover, chronic treatments with E2 and-or OXT may roughly simulate the *in vivo* condition in which animals treated with E2 shown a strong increase of plasmatic OXT (*Divari et al., 2013*). The use of OXT in the chronic treatment, despite OXT can be regulated by E2, was considered necessary for several reasons: (a) the OXT secretion induced by E2 is very small and a hypothetical effect on muscle cells would have been very hard to identify, (b) the use of a certain OXT concentration for treatment would allow authors to clearly establish a causal relationship between treatment and effect. To elicit the time points for chronic treatments we considered that E2 half-life is estimated to be around 24 h (*Strobl & Lippman, 1979*) but a daily medium change could have damaged the cells. Despite OXT short half-life *in vivo* (1–2 min in blood and 28 min in cerebrospinal fluid) (*Gimpl & Fahrenholz, 2001*), other authors (*Cassoni et al., 2002*) already performed treatments with OXT and/or E2 for 48 h in cell culture.

Chronic treatments with E2 and/or OXT did not induce significant morphological modifications of MTs. This result does not confirm literature which provides few contradictory data. *Kiss & Mikkelsen (2005)* described in E2-treated C2C12 MTs a decrease

of the fusion index by approximately 50% versus the control. On the other side, our result about *Myh* downregulation is similar to those reported in Ogawa's work (*Ogawa et al., 2011*) where a significant decrease of the levels of the MYH and MYOG proteins was described in E2-treated murine satellite cells. MYH is the dominant myofibrillar protein in differentiated muscle and a downregulation would occur during an inhibitory process. On the contrary, other authors described that E2 promotes differentiation of rat MB cell line and induces changes of some differentiation markers such as myogenin and creatine kinase (*Galluzzo et al., 2009*; *Ronda & Boland, 2016*). Unfortunately, a large heterogeneity exists in literature which includes the use of different cell lines and the choice of different time points for treatment and evaluation, making difficult to take a definitive conclusion about E2 effect on muscular differentiation.

A further reflection could be carried out on the modulation of lipid metabolism in the muscle. Reduction of FABP4 levels has been associated with a better response to insulin and protective effect against obesity (*Makowski & Hotamisligil, 2004*). In our study, *Fabp4* gene was mildly downregulated by the treatment with OXT only, suggesting a positive effect of OXT on glucose and lipid metabolism in skeletal muscle, as previously reported (*Elabd & Sabry, 2015*). On the contrary, a mild overexpression of *Fabp4* was described after the administration of the combination of E2 and OXT and this data might be related to a negative influence of high E2 doses on glucose incorporation by MTs, as described by *Garrido et al. (2014)*, but the responsible mechanisms and the meaning are still unclear.

In conclusion, the detection of a mild but significant downregulation of *Myh* may lead to the hypothesis that *in vitro* muscular atrophy might be associated with estrogen-induced glucose and lipid dysmetabolism. The concomitant increase in *Oxtr* gene expression and the mild secretion of OXT peptide may be considered as an attempt to restrain the catabolic process and support glucose uptake. *Altszuler & Hampshire (1981)* and *Lee et al. (2008)* described that OXT is able to induce glucose uptake and increases the plasma insulin levels. Moreover, *Elabd et al. (2014)* demonstrated that OXT regulates muscle maintenance and repair in mice. For these reasons, it is likely that the OXT/OXTR system is involved in skeletal muscle metabolism and development, but further investigation are needed in order to better understand the multiple metabolic activities of OXT in skeletal muscle.

## CONCLUSIONS

The study demonstrates that the expression of OXT and OXTR increases with the differentiation process in C2C12 cell line. E2 upregulates OXT and OXTR in C2C12 MTs, similar to the *in vivo* experiments performed on cattle treated with hormones. This cell line model could be used for subsequent studies to understand the role of the OXT peptide on skeletal muscle development and metabolism in relation to E2 administration. In particular, the non-genomic and genomic mechanisms by which E2 regulates OXT synthesis/release are of great interest as well as the effects of these two hormones on glucose and lipid homeostasis.

Since skeletal muscle has been recognized as a secretory organ, OXT could work via autocrine or paracrine mechanisms to regulate skeletal muscle metabolism. Further studies

are needed to increase our knowledge about estrogen influence on muscle and its interaction with OXT.

## ACKNOWLEDGEMENTS

The authors are grateful to Marta Leporati for providing scientific support, Domenico Palmerini and Alessandra Sereno for supplying technical support and to the ''Bruno Maria Zaini'' Reference Centre of Comparative Pathology, Department of Veterinary Science, University of Turin, Italy.

### Funding

This work was partially funded by the University of Turin, 2012 project ''Valutazione di test alternativi ai test ufficiali per l'identificazione di vitelli trattati con promotori di crescita''. There was no additional external funding received for this study. The funders had no role in study design, data collection and analysis, decision to publish, or preparation of the manuscript.

### Grant Disclosures

The following grant information was disclosed by the authors:
University of Turin.

### Competing Interests

The authors declare there are no competing interests.

### Author Contributions

- Enrica Berio conceived and designed the experiments, performed the experiments, wrote the paper, prepared figures and/or tables.
- Sara Divari performed the experiments, wrote the paper.
- Laura Starvaggi Cucuzza analyzed the data.
- Bartolomeo Biolatti contributed reagents/materials/analysis tools, reviewed drafts of the paper.
- Francesca Tiziana Cannizzo conceived and designed the experiments, contributed reagents/materials/analysis tools, reviewed drafts of the paper.

### Data Availability

  The raw data has been supplied as a Supplementary File.

### Supplemental Information

Supplemental information for this article can be found online at http://dx.doi.org/10.7717/peerj.3124#supplemental-information.

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
