# Peer review of "β-estradiol upregulates oxytocin and the oxytocin receptor in C2C12 myotubes"

_PeerJ, doi:10.7717/peerj.3124_

## Round 0.1 · original submission · Major Revisions

Please consider all the reviewer suggestions in the revised manuscript. Data presentation and figures have to be improved as indicated.

Reviewer 1 ·

Basic reporting

The study is novel but the experiments need to include controls and quantification.
Additionally there are several punctuation errors throughout the text as well as some grammatical errors in the methods section

Experimental design

Kindly explain why the various time points were chosen for the study. For eg: 8 days for chronic and 0.5, 1, 2, 3, and 4 hours time points for acute treatments. In the introduction, add information about the protein degradation profile of Oxt and E2 which is very important in deciding treatment time points (in-vitro).

Define myotube fusion index. Add representative images in main figures for better understanding.

Validity of the findings

Table 2: convert it into a graph for better representation.

Figure 1: add "days" next to the number of days in the figure. In acute treatment why was only E2 treatment added and external Oxt treatment avoided which is present in chronic treatment. I think it would be very interesting to see the effect in acute for both (E2+Oxt).

Figure 2: Include asterix to indicate the significance in the graph for RT-PCR.
2B) There is no marker in the western blot. Also, a positive control should be included to validate the specificity of the OXTR antibody. Kindly include the number of repeats for western blots and quantify using ImageJ.

Figure 3: Include positive control for B and C. Fig 2: Quantify using imageJ

Fig 4: include representative images.

One general query: Other than Figure 3A, there is no statistical significant difference between any of the treatments? The figures suggest that.

Additional comments

See above comments.

Reviewer 2 ·

Basic reporting

No comments

Experimental design

No Comments

Validity of the findings

No comments

Additional comments

The manuscript “17β-Estradiol Up-regulates Oxytocin and the Oxytocin Receptor in C2C12 Myotubes.” by Berio and colleagues studies the modulations of 17β-estradiol affect oxytocin, glucose and lipid metabolism in C2C12 myoblasts and myotubes. The study will be of interest to the wide audience, such as endocrinology, sports medicine communities, but there are several concerns that need to be addressed before publication.

1. Both previous studies and acute treatment experiments in this manuscript demonstrate that E2 can induce OXT expression in skeletal muscle, which suggests the expression of OXT is E2-dependent. Why did the authors choose E2, OXT, E2+OXT as the chronic treatment groups? Theoretically, the three groups are not independent.

2. For figure 4, the authors compared the fusion index of chronic treatment groups and drew the conclusion that not OXT but E2, E2+OXT can decrease the fusion index. However, taking into account the relatively big error bars in this figure, it’s hard to see whether there are differences among these fusion indexes. The authors should rewrite related conclusions.

3. For the chronic treatment, the authors mentioned that they treated cells every 48 hours. But the arrows in figure 1 indicate the administration times are day 1, 3, 6, 8. The authors need double check the time table.

4. In line 259, the authors claimed that the disagreements between their results and previous reports about chronic treatments are caused by different methods. Could the authors give more comparisons and discussions here so that reader can have a better understanding for these methods?

5. The manuscript is clearly written, logically organized, methods are well explained. However, the quality of writing needs to be improved further. Redundant or ill-used ‘a’ spread all over the manuscript, especially in the “material and methods”, such as ‘a 5% CO2’ in line 92, ‘a the Mini Trans-Blot cell’ in line 152. Other issues include: line 24, what’s the meaning of “they”? Line 64, ‘perproprotein’ should be ‘preproprotein’.

Reviewer 3 ·

Basic reporting

No Comments

Experimental design

No Comments

Validity of the findings

No Comments

Additional comments

Berio et al has investigated the relationship of 17β-estradiol on skeletal muscle cell C2C12. They demonstrated that E2 treatment upregulates Oxt expression by Real-time PCR, Western-blot and ELISA. However, there remain some defects in this work.
1. In Figure 2, the mRNA and protein level of Oxt should also be shown.
2. In Figure 3, Oxt mRNA was up regulated by approximately 3-fold after 0.5 hours of E2 treatment, while in table 2, E2 treatment has no effect on Oxt mRNA expression, why? In addition, the expression profiles of Oxtr should also be shown.
3. In Figure 3, in order to make the reader better understand the effect of chronic E2 treatment on myogenic development, the representive H&E staining picture should be given.

---

## Round 0.2 · accepted · Accept

Congratulations for your work which now can be published.

Reviewer 2 ·

Basic reporting

no comment

Experimental design

no comment

Validity of the findings

no comment

Additional comments

The authors have addressed the reviewers’ concerns and made the appropriate changes accordingly. I recommend that this paper be accepted for publication.

Reviewer 3 ·

Basic reporting

No comment

Experimental design

No comment

Validity of the findings

No comment

Additional comments

The authors have answered most of my queries satisfactorily and added additional data for clarification, and have greatly improved the manuscript.